# PPARδ and FOXO1 Mediate Palmitate-Induced Inhibition of Muscle Pyruvate Dehydrogenase Complex and CHO Oxidation, Events Reversed by Electrical Pulse Stimulation

**DOI:** 10.3390/ijms21165942

**Published:** 2020-08-18

**Authors:** Hung-Che Chien, Paul L. Greenhaff, Dumitru Constantin-Teodosiu

**Affiliations:** 1School of Life Sciences, University of Nottingham, Queen’s Medical Centre, Nottingham NG7 2UH, UK; hung.chien@nottingham.ac.uk (H.-C.C.); paul.greenhaff@nottingham.ac.uk (P.L.G.); 2Department of Physiology and Biophysics, National Defense Medical Centre, Taipei 11490, Taiwan

**Keywords:** free fatty acids, insulin resistance, muscle cell, contraction

## Abstract

The mechanisms behind the reduction in muscle pyruvate dehydrogenase complex (PDC)-controlled carbohydrate (CHO) oxidation during chronic high-fat dietary intake are poorly understood, as is the basis of CHO oxidation restoration during muscle contraction. C2C12 myotubes were treated with (300 μM) palmitate or without (control) for 16 h in the presence and absence of electrical pulse stimulation (EPS, 11.5 V, 1 Hz, 2 ms). Compared to control, palmitate reduced cell glucose uptake (*p <* 0.05), PDC activity (*p <* 0.01), acetylcarnitine accumulation (*p <* 0.05) and glucose-derived mitochondrial ATP production (*p <* 0.01) and increased pyruvate dehydrogenase kinase isoform 4 (PDK4) (*p <* 0.01), peroxisome proliferator-activated receptor alpha (PPARα) (*p <* 0.01) and peroxisome proliferator-activated receptor delta (PPARδ) (*p <* 0.01) proteins, and reduced the whole-cell *p*-FOXO1/t-FOXO1 (Forkhead Box O1) ratio (*p <* 0.01). EPS rescued palmitate-induced inhibition of CHO oxidation, reflected by increased glucose uptake (*p <* 0.01), PDC activity (*p <* 0.01) and glucose-derived mitochondrial ATP production (*p <* 0.01) compared to palmitate alone. EPS was also associated with less PDK4 (*p <* 0.01) and PPARδ (*p <* 0.01) proteins, and lower nuclear *p*-FOXO1/t-FOXO1 ratio normalised to the cytoplasmic ratio, but with no changes in PPARα protein. Collectively, these data suggest PPARδ, and FOXO1 transcription factors increased PDK4 protein in the presence of palmitate, which limited PDC activity and flux, and blunted CHO oxidation and glucose uptake. Conversely, EPS rescued these metabolic events by modulating the same transcription factors.

## 1. Introduction

Obesity and chronically elevated levels of circulating free fatty acids (FFA) are recognised as factors contributing to the development of metabolic inflexibility (MI; defined as the inability to switch from lipid to glucose oxidation during the fasting-to-fed transition), insulin resistance (IR), and type 2 diabetes mellitus [1]. Metabolic inflexibility is also accompanied by impairment of muscle mitochondrial function and fuel use [2,3]. Skeletal muscle is the primary site of whole-body oxidative (mitochondrial) and non-oxidative (glycolytic) glucose disposal, and glucose accretion in the form of glycogen, and therefore is a major contributor to MI and IR under pathophysiological conditions [4,5]. Palmitate, the most common saturated fatty acid in the diet, is often used to induce cellular MI and IR in vitro and in vivo by blunting insulin signalling and decreasing cellular glucose transport and mitochondrial carbohydrate (CHO) use [6,7,8,9]. Central to this latter response is the evidence that chronically elevated palmitate directly inhibits the activity status of the pyruvate dehydrogenase complex (PDC) activity, which is the rate-limiting step in mitochondrial CHO oxidation [10].

The activity of PDC is controlled primarily by a covalent mechanism involving two competing pyruvate dehydrogenase kinase (PDK) and phosphatase (PDP) reactions [11]. The PDK family is comprised of four isoforms (PDK1-4), whereas PDP consists of two isoforms (PDP1-2). PDK2 and PDK4 mRNA are specifically expressed in skeletal muscle, but since the specific activity of PDK4 is eightfold higher than that of PDK2 [12], PDK4 has been assigned greater regulatory significance in muscle. Indeed, after high-fat dietary intake, and in starvation and diabetes, the expression of PDK4 mRNA and protein are consistently increased in most tissues in rodents [13] and humans [14], and accordingly the activity of PDC is reduced. Furthermore, muscle PDC activity is most potently increased during contraction, which is mediated by intra-mitochondrial calcium accumulation, which activates PDP1 [10].

In vitro studies [11] have suggested that in addition to the translation (covalent, i.e., amount of protein) regulation of PDK4 by palmitate, there is potentially a second post-translational (allosteric; enzymatic activation/inhibition) regulation of PDK4 activity through changes in NADH/NAD^+^, acetyl-CoA/CoASH, ATP/ADP ratios or pyruvate availability. However, in vivo studies involving voluntary incremental or exhaustive exercise or involuntary high-intensity electrically evoked muscle contraction, have shown near-maximal activation of the PDK4 downstream target PDC in the presence of marked increases in the cellular NADH/NAD^+^ and acetyl-CoA/CoASH ratio [15,16,17]. Full activation of PDC by high-intensity exercise occurs in human skeletal muscle even during conditions of glycogen depletion, and therefore low pyruvate availability, or near-full reduction of cellular NAD^+^ to NADH by prolonged occlusion of the blood supply the lower limb [18]. Collectively, these studies strongly advocate exercise-mediated calcium release is the main driver of PDC activation in the face of high ratios of acetyl-CoA/CoASH and NADH/NAD^+^ [10].

Peroxisome proliferator-activated receptors (PPARs) are a group of nuclear receptor proteins that function as transcription factors regulating the expression of several genes that control several cellular processes like inflammation, adipogenesis, lipid metabolism, glucose homeostasis, and insulin resistance [19,20,21]. PPARs can be classified into three isotypes: α, β/δ, and γ, which display differential tissue distribution and specific functions [22]. As activation of PPARs with natural (free fatty acids) [23,24] or synthetic [25] ligands have been associated with increases in muscle PDK4 mRNA and protein expression, which would thereby inhibit PDC activity, it is, therefore, pertinent to consider that PPARs activation is, in fact, a trigger for MI.

FOXO1 is a transcription factor expressed mostly in skeletal muscle, liver and adipose tissue [26]. Free fatty acids can indirectly induce the translocation of FOXO1 to the nucleus [27], and evidence has shown FOXO1 can directly increase PDK4 mRNA expression by binding to the promoter region of the PDK4 gene [27]. It is therefore plausible that FOXO1 could also play an important role in controlling muscle PDK4 expression and thereby reducing PDC flux and CHO oxidation in response to increased FFA availability.

Isolated bouts of voluntary and electrically evoked muscle contraction fully activate muscle PDC in humans [16,17]. This response is blunted when exercise is preceded by three days of high-fat dietary intake, irrespective of the type or intensity of muscle contraction performed [14,28]. Electrical pulse stimulation (EPS) of cultured muscle cells is also a useful model to mimic muscle contraction in humans, and can be sustained for prolonged periods and avoids central fatigue [29]. Electrical pulse stimulation of cultured muscle cells is also known to increase cellular glucose uptake [30] and CHO oxidation [31]. However, whether chronic use of EPS can reverse the palmitate-induced PDC inhibition and increase mitochondrial CHO use is unknown.

We hypothesised here that palmitate administration would reduce glucose uptake, PDC activity and flux and CHO use in C2C12 myotubes compared to control. Furthermore, these events would be associated with increased peroxisome proliferator-activated receptor alpha and delta (PPARα and δ) and FOXO1 transcription factor expression relative to control, and accordingly increased muscle PDK4 protein content. Finally, we also hypothesised that EPS would rescue palmitate-induced inhibition of PDC activity and CHO use in C2C12 myotubes by modulating these same molecular and cellular events.

## 2. Results

### 2.1. Viability of C2C12 Myotubes with Increasing Palmitate Concentrations

To identify the maximal palmitate concentration that would allow full cell sustainability, the viability of C2C12 myotubes with increasing palmitate concentrations was determined with the MTT 3-(4,5-dimethylthiazol-2-yl)-2,5-diphenyltetrazolium bromide assay. Figure 1 shows that there was no significant change in cell viability in the palmitate concentration range of 0–500 µM, albeit a non-significant decline at 500 μM was noticeable. However, C2C12 myotubes treated with 750 µM palmitate showed a clear decline in viability. Based on this preliminary investigation, we choose 300 μM as the working palmitate concentration.

### 2.2. Palmitate Reduced C2C12 Myotube Glucose Uptake, an Event Which Was Reversed When Combined with EPS Intervention

The effects of palmitate (PAL), EPS or the combination of PAL + EPS on glucose uptake and cell media lactate concentrations in C2C12 myotubes are shown in Figure 2A (glucose) and Figure 2B (lactate). Compared to control, palmitate alone reduced C2C12 myotube glucose uptake (88.0 ± 3.6%; *p <* 0.05; Figure 2A), while EPS alone increased it (139.0 ± 6.3%, *p <* 0.01). Electrical pulse stimulation rescued the palmitate-induced inhibition of glucose uptake (119.0 ± 7.7%, *p <* 0.01; Figure 2A). Cell media lactate concentrations during the same experimental conditions are presented in Figure 2B. Administration of both EPS and palmitate increased medium lactate concentration compared to control (both *p <* 0.01). The combination of palmitate and EPS further increased the medium lactate concentration over control (*p <* 0.01), and also above EPS and palmitate (both *p <* 0.05).

### 2.3. Palmitate Reduced C2C12 Myotube PDC Activity and Acetylcarnitine Concentration, Events Which Were Reversed When Combined with EPS Intervention

The effects of palmitate (PAL), EPS or the combination of PAL + EPS on PDC activity and acetylcarnitine concentrations in C2C12 myotubes are shown in Figure 3A (PDC) and Figure 3B (acetylcarnitine). Palmitate reduced PDC activity compared to control (3.0 ± 0.2 and 3.7 ± 0.2 pmol acetyl-CoA/min/mg protein, *p <* 0.01; Figure 3A), while EPS alone increased it above control (4.6 ± 0.2 pmol acetyl-CoA/min/mg protein, *p <* 0.01). This palmitate-induced inhibition of PDC activity was rescued by EPS (3.6 ± 0.2 pmol acetyl-CoA/min/mg protein), such that there was no difference from control, but activity was less than EPS alone (*p <* 0.01). Myotube acetylcarnitine content was determined as an index of PDC flux (Figure 3B). Palmitate reduced acetylcarnitine content compared to control (105 ± 11 vs. 83 ± 4 pmol/mg protein, *p <* 0.05), while EPS alone increased it (172 ± 30 pmol/mg protein, *p <* 0.01). Moreover, EPS rescued the palmitate-induced reduction in acetylcarnitine content (142 ± 3 pmol/mg protein, *p <* 0.01).

### 2.4. Palmitate Increased C2C12 Myotube PDK4 Protein Expression with Palmitate Treatment, an Event Which Was Reversed When Combined with EPS Intervention

The effects of palmitate (PAL), EPS or the combination of PAL + EPS on PDK 4 and α-actin protein expressions in C2C12 myotubes are shown in Figure 4A (treatment group protein bands) and Figure 4B (mean bands intensities). Palmitate alone increased C2C12 myotube PDK4 protein expression 2.5-fold relative to control (*p <* 0.01; Figure 4B). Electrical pulse stimulation alone did not affect myotube PDK4 protein expression compared to control, but almost normalised the increase observed with palmitate alone (1.4 ± 0.1 vs. 2.5 ± 0.1, *p <* 0.01), such that the protein expression following combined administration of palmitate and EPS was no different from control. There was also a significant negative correlation (*p* = 0.002; *r^2^* = −0.603) between individual PDK4 protein expression (Figure 4B) and the corresponding activity of PDC in its dephosphorylated active form (PDCa) (Figure 3A).

### 2.5. Palmitate Reduced C2C12 Myotube Maximal Mitochondrial ATP Production Rates from Pyruvate + Malate, an Event Which Was Reversed When Combined with EPS Intervention

The effects of palmitate (PAL), EPS or the combination of PAL + EPS on maximal glucose-derived pyruvate mitochondrial ATP production rates (MAPR) in C2C12 myotubes are shown in Figure 5. Palmitate reduced MAPR compared to control (9.8 ± 0.3 vs. 10.8 ± 0.2 nmol ATP/min/mg protein, *p <* 0.01; Figure 5), while EPS increased it (12.0 ± 0.3 nmol ATP/min/mg protein, *p <* 0.01). Electrical pulse stimulation rescued the palmitate-induced reduction in MAPR (10.8 ± 0.3 nmol ATP/min/mg protein, *p <* 0.01).

### 2.6. Palmitate Increased C2C12 Myotube PPARα and PPARδ and Reduced the Ratio of p-FOXO1/t-FOXO1 Protein Expressions in Total Cellular Lysates. Addition of EPS Intervention Reverted the Palmitate Mediated Upregulation of PPARδ Protein

Western blots depicting PPARα, PPAδ, phosphorylated (*p*)-FOXO1, total (t)-FOXO1 and α-actin protein bands with their corresponding molecular weights in response to palmitate (PAL), EPS or the combination of PAL + EPS are shown in Figure 6A. Their mean band intensities normalised to α-actin are presented in Figure 6B–D, respectively. Palmitate increased myotube PPARα and PPARδ protein expression (1.5 ± 0.1 and 1.9 ± 0.1 fold over control; *p <* 0.01 and *p <* 0.01, respectively; Figure 6B,C) and reduced the *p*-FOXO1/t- FOXO1 protein ratio relative to control (0.40 ± 0.02, *p <* 0.01; Figure 6D). Electrical pulse stimulation had no further impact on PPARα and PPARδ protein expression levels compared to control but reduced the palmitate-induced upregulation of PPARδ protein expression (*p <* 0.01; Figure 6C). There was no further impact of EPS + PAL on PPARα or the *p*-FOXO1/t-FOXO ratio when compared to palmitate alone (Figure 6B,D).

### 2.7. Electrical Pulse Stimulation with or without Palmitate Significantly Reduced the p-FOXO1/t-FOXO1 Ratio Normalised to that of Nuclear to Cytoplasmic Factor

Typical Western blots depicting phosphorylated (*p*)-FOXO1 and total (t)-FOXO1 protein bands with their corresponding molecular weights in the nuclear and cytosolic fractions in response to palmitate (PAL), EPS or the combination of PAL + EPS are shown in Figure 7A (nuclear) and Figure 7B (cytosolic). The mean band intensities of the phosphorylated to total ratios in all groups are presented in Figure 7C. Palmitate had no impact on the *p*-FOXO1/t-FOXO1 ratio normalised to that of nuclear to cytoplasmic factor. However, EPS reduced this ratio in the absence (57%, *p <* 0.01) and the presence (54%, *p <* 0.01) of palmitate (Figure 7C).

## 3. Discussion

The present study demonstrates that treatment of C2C12 myotubes with a palmitate concentration within physiological ranges (300 μM) reduces the cellular glucose uptake, PDC activity and flux, and rates of glucose-derived mitochondrial ATP production (MAPR) compared to control (no palmitate). Furthermore, these palmitate-induced cellular events were accompanied by increased PPARα (50%) and PPARδ (200%) protein expression and a reduction in the *p*-FOXO/t-FOXO protein ratio (60%) in total cell lysates, alongside higher PDK4 protein expression relative to control. Of note, exposure of palmitate-treated cells to EPS rescued this palmitate-induced dysregulation of CHO metabolism, as reflected by the restoration of deficits in cellular glucose uptake, PDC activity and its flux, and the mitochondrial glucose-derived pyruvate ATP production. Furthermore, these events were accompanied by the normalisation of the protein expression levels of PDK4, PPARδ, but not PPARα, and the *p*-FOXO1/t-FOXO1 ratio normalised to that of nuclear to cytoplasmic factor. Overall, therefore, these novel data suggest that palmitate-induced increases in the expression of PPARδ and FOXO1 transcription factors underpin the downstream increase in myotube PDK4 protein expression, which then limits PDC activity and flux, and blunts CHO oxidation and cellular glucose uptake. Furthermore, the application of EPS blunts this palmitate-induced dysregulation of CHO metabolism via modulation of the same molecular and cellular events.

A consistent finding in the literature is the decline in muscle PDC activity in tandem with the increase in muscle PDK4 protein expression when muscle is exposed to changes in FFA and insulin concentrations, such as during fasting, pathophysiological conditions (e.g., type 2 diabetes), or during pharmacological intervention with, for example, peroxisome proliferator-activated receptor (PPAR) agonists or statins [10,23,24,32,33,34,35]. Previous studies involving human volunteers have shown a direct relationship between the intensity of muscle contraction and the magnitude of PDC activation, and thereby the rate of muscle glucose oxidation [17,28,34]. These in vivo studies also showed that full activation of muscle PDC was achieved within seconds of the onset of electrically evoked high-intensity isometric contraction, although this was not the case if the exercise was preceded by three days of dietary high-fat intake [14]. The present study was able to show, however, that in vitro 16 h of pulse electrical stimulation rescued PDC activity and flux, CHO oxidation and cellular glucose uptake from the metabolic effects of palmitate. Likely explanations for the difference between the in vivo [14] and current in vitro observations lies in differences in the duration and intensity of muscle contraction employed. Certainly, it is reasonable to conclude that 16 h of contraction is far more likely to elicit molecular changes at a protein level than short bouts of contraction.

Electrical pulse stimulation of differentiated skeletal muscle cells (myotubes) in culture has been proposed to replace the motor neuron junction (MNJ) activation of muscle fibres [29]. However, since EPS excites the muscle cell directly via the cellular membrane, a pertinent question would be what is the status of calcium (Ca^2+^) ions? Would they be made available to activate PDC? Usually, upon MNJ activation, Ca^2+^ would be released from the sarcoplasmic reticulum (ER), which surrounds each myofibril, to flood the muscle cell. The ER contains ATP-driven Ca^2+^ pumps, which would then generate a high Ca^2+^ concentration in the ER lumen. When inositol 1,4,5-trisphosphate (IP3) binds to IP3 receptors, the channel region of the receptor opens, allowing Ca^2+^ to flood out into the cytosol. At the same time, the Ca^2+^ uptake by mitochondria via the mitochondrial transmembrane protein calcium uniporter would facilitate activation of PDP1 [10], and thereby the dephosphorylation/activation of PDC. Overall, it was reassuring when it was demonstrated that EPS, similar to MNJ activation, can raise both IP3 and calcium ions in rat myocytes [36].

It is worth considering in more detail that the normalisation of palmitate-induced deficits in cellular glucose uptake, PDC activity and flux, and rates of CHO-derived MAPR after 16 h of EPS were matched by reductions in PDK4, PPARδ, and the nuclear to cytosolic *p*-FOXO1/t-FOXO1 ratio (Figure 6 and Figure 7) protein expression, but not in a reduction in PPARα protein expression.

Of note, the *p*-FOXO1/t-FOXO1 ratios in total muscle lysate with palmitate were similar with or without EPS intervention (Figure 6D). However, when considering the nuclear to cytosolic *p*-FOXO1/t-FOXO1 ratios with EPS with or without palmitate, EPS significantly reduced the availability of dephosphorylated FOXO1 (active form) in the nucleus (Figure 7C). Here, a pertinent question is what the upstream regulator of *p*-FOXO1/t-FOXO1 protein ratio might be? While palmitate as a natural ligand of PPARs can activate the expression of PPARs directly, we showed previously that the change in *p*-Akt1/t-Akt1 (serine/threonine-protein kinase 1) protein ratio seems to be one of the upstream regulators of *p*-FOXO1/t-FOXO1 protein ratio [37]. Thus, in this in vivo porcine model, the outcomes of the *p*-FOXO1/t-FOXO1 protein ratios in response to various systemic circulating FFAs levels induced by a starvation-feeding cycle were tightly and positively correlated to changes in *p*-Akt1/t-Akt1 (*r*^2^ = 0.60) [37]. Related to the effect of EPS on Akt1, the stimulation of muscle Akt1 phosphorylation during contractile activity is well known [38].

Collectively, these findings suggest that PPARδ and FOXO1 transcription factors are important in controlling the expression of PDK4 protein. In the case of FOXO1, dephosphorylation (activation) results in its translocation from the cytosol to the nucleus, where it can bind directly to the promoter region of the PDK4 gene [27]. Moreover, FOXO1 dephosphorylation appears to be coupled via a signalling cascade to changes in circulatory FFA and/or insulin availability [39], thereby tying muscle PDK4 gene transcription (amongst others) to substrate availability. Indeed, we have previously highlighted a role for FOXO1 in the up-regulation of muscle PDK4 mRNA expression, and reduction in muscle PDC activity, in the resting state and during 60 min of submaximal exercise following 3 days of dietary high-fat feeding in healthy volunteers [14].

Myotube acetylcarnitine content was presently determined to advance our understanding of whether the intermediary metabolism responses to the study interventions match the molecular changes. There is usually a positive relationship between the activation of PDC flux and intracellular levels acetylcarnitine accumulation as when the rate of PDC mediated mitochondrial acetyl-CoA formation exceeds the rate of acetyl-CoA entry into the Krebs cycle, cellular acetylcarnitine content increases [16]. However, in the presence of palmitate, the cellular acetylcarnitine accumulation was significantly lower than in control and EPS groups indicative of a lower PDC flux (Figure 3A,B). Nevertheless, when EPS was applied in the presence of palmitate, the levels of PDC flux and acetylcarnitine were restored.

It is worth noting that glucose uptake was less and lactate accumulation greater in the PAL group than in control (Figure 2A,B). Although this may seem counterintuitive, the lower glucose uptake in the presence of palmitate was also associated with lower PDCa (Figure 3A) and therefore less PDC flux (Figure 3B), which secondary to this contributed to an increase in lactate formation from pyruvate. These findings bear similarities to a recent in vivo report where a bout of maximal intense exercise was unable to overcome dietary fat-mediated inhibition of muscle pyruvate dehydrogenase complex activation and was associated with greater muscle lactate accumulation, as a result of lower PDC flux [28].

Considering the role of PPARδ in the context of the current findings, we have previously reported in vivo and ex-vivo rat model studies that pre-treatment with the PPARδ agonist compound (GW610742) increased muscle PDK4 protein expression [25] and attenuated exercise-induced activation of muscle PDC and MAPR [34]. Conversely, work by others revealed that PPARδ knock-out in mice was associated with a decline in PDK4 mRNA expression [40]. Collectively, these observations point to an important role for PPARδ in the regulation of muscle PDC activation and flux and mitochondrial CHO oxidation. Although PPARα, alone or in association with PPARδ, was previously also considered as a regulator of muscle PDC activity during conditions associated with increased FFA availability (e.g., starvation or dietary high-fat intake) [21,23], the present study revealed no clear association between changes in cellular fuel metabolism and PPARα protein expression.

In conclusion, the present results point to potentially important roles for PPARδ and FOXO1 in the palmitate-induced increase in myotube PDK4 expression and linked reductions in cellular glucose uptake, PDC activity and flux, and rates of CHO-derived oxidative ATP generation. Furthermore, the finding that 16 h of EPS rescued this palmitate-induced dysregulation of cellular CHO metabolism via modulation of the same molecular and cellular events summarised in Figure 8, alludes to the mechanisms by which chronic physical activity will protect skeletal muscle against lipid-induced muscle metabolic inflexibility and IR.

## 4. Materials and Methods

### 4.1. Materials

All cell culture media (Dulbecco’s modified Eagle’s medium, phosphate-buffered saline, and penicillin-streptomycin), sodium palmitate, sodium pyruvate, 2-deoxy-D-glucose were purchased from Sigma Aldrich (St. Louis, MO, USA). Foetal bovine serum (FBS), horse serum, cell extraction buffer and cocktail inhibitor were obtained from Life Technologies (Warrington, UK). Tritiated glucose, D-[3-3H] was obtained from Perkin Elmer (Boston, MA, USA). Mitochondrial ATP monitoring reagent was purchased from Biothema (Handen, Sweden). Protein assay reagents were purchased from BioRad (Copenhagen, Denmark). Anti-PDK4 (ab214938), anti-PPAR alpha (ab24509), anti-PPAR delta (ab23673), goat anti-Mouse (IRDye^®^ 800CW) (ab216772), goat Anti-Rabbit (IRDye^®^ 680RD) (ab216777) and anti-alpha actin (ab28052) antibodies were purchased from Abcam (Cambridge, UK). Anti-Phospho-FoxO1 (Ser256) (#9461) and Anti-FoxO1 (#2880) antibodies were purchased from Cell Signaling Technology (London, UK). Levels of cytosolic and nuclear fractions of *p*- and t-total FOXO1 were identified using the same antibodies that were used for total cellular lysate measurements but were applied to isolated cellular fractions using a nuclear protein extraction kit (Active Motif Europe, La Hulpe, Belgium).

### 4.2. Cell Culture

C2C12 skeletal muscle cells were purchased from ATCC (Manassas, VA, USA), cultured in collagen-coated 6-well plates, maintained in a growth media containing High Glucose Dulbecco’s Modified Eagle Media (DMEM), 10% Foetal Bovine Serum (FBS) and a 1% Pen/Strep. Once 80–100% confluent, the cells were differentiated using a differentiating media containing high glucose DMEM, 2% Donor Horse Serum and 1% Pen/Strep. Cells were ready for stimulation after C2C12 myotubes were fully differentiated into contractile myotubes (4 to 5 days).

To test our hypotheses, C2C12 myotubes were investigated in four experimental interventions: after exposure to either standard growth media (CON); 16 h of 300 μM palmitate (PAL); 16 h of electrical pulse stimulation (EPS) or the combination of both PAL and EPS interventions (PAL + EPS).

### 4.3. Electrical Pulse Stimulation

Fully differentiated C2C12 myotubes were induced to contract by EPS. The stimulation pulse was set at 2 ms, delivering 11.5 V at a frequency of 1 Hz for 16 h using C-Dish combined with a pulse generator (C-Pace 100; IonOptix, Milton, MA, USA). This instrument delivered electrical stimuli via the carbon electrodes of the C-dish, which were placed into the culture medium. Cell extracts from each dish were prepared immediately after termination of EPS.

### 4.4. Glucose Uptake Assay

Glucose uptake was determined, as previously reported [41]. Briefly, C2C12 myotubes attached to the plate were incubated in 1 mL of media buffer supplemented with 138 mM NaCl, 1.85 mM CaCl2 1.3 mM MgSO4, 4.8 mM KCl, 50 mM HEPES pH 7.4 and 0.2% (W/V) BSA for 2 h. Then, 250 mL of 2-deoxy-H^3^-glucose (1 mCi/mL) was added to the well, and the cells were incubated for 15 min. After incubation, the media were removed, and the well was washed three times with cold PBS buffer. After the removal of the last wash content, 500 mL of NaOH 50 mM and SDS 0.1% were added to solubilise the cells. An aliquot of the solubilised cell mixture was removed and pipetted into a vial containing 5 mL of scintillation liquid, and the radioactivity was measured with a scintillation counter.

### 4.5. Cell Metabolite Levels and PDC Activity Measurements

Medium lactate concentration was determined fluorometrically using a modified spectrophotometric method [42]. Acetylcarnitine (as an index of PDC flux; [16]) was measured in cell extract using an enzymatic radioactive assay as previously described [43]. PDC activity was measured as described previously [10]. Briefly, C2C12 cells were extracted in a buffer containing sodium fluoride (NaF) and dichloroacetate (DCA). The activity of PDC in its dephosphorylated active form (PDCa) was assayed and expressed as the rate of acetyl-CoA formation (pmol acetyl-CoA/min/mg protein) at 37 °C.

### 4.6. Determination of Maximal Mitochondrial ATP Production Rates in Intact C2C12 Myotubes

The maximal mitochondrial ATP production rates (MAPR) in C2C12 myotubes were measured using a method described previously [44]. Of note, the MAPR measurements were conducted in the absence of palmitate, which was removed after cell pelleting and resuspension in the cell permeabilisation buffer, followed by resuspension in the ATP measuring buffer. Briefly, 3 μL of permeabilised cells was added to each well of a 96-well luminometer plate. Each well contained 200 µL ATP monitoring reagent (firefly luciferase, 357 µM D-luciferin, 14.3 µM L-luciferin, 15 mM BSA, 1 µM sodium pyrophosphate decahydrate, 186 mM sucrose, 18.8 mM monopotassium phosphate, 2.5 mM magnesium acetate tetrahydrate, 677 µM K_2_EDTA pH 7.0), 12.5 µL of 12 mM ADP and 35 µL of 375 mM pyruvate + 143 mM malate. Luminescence was recorded continuously for 10 min (BMG LABTECH Ltd., Ortenberg, Germany), after which an injection of 150 pmoles ATP standard took place. The change in luminescence elicited by the ATP standard was used to calculate values for absolute MAPR.

### 4.7. Western Blotting

Near-Infrared (NIR) Western Blot Detection (Li-cor Biosciences UK Ltd Odyssey CLx, Cambridge, UK) which is more dynamic, covers a wider linear range, and has a higher sensitivity than the traditional chemiluminescent Western blotting was employed. Briefly, C2C12 myotubes were lysed with a buffer containing a cocktail of protease and phosphatase inhibitors. The total protein concentration was determined using a protein assay (Bio-Rad, Watford, UK). In total, 25 μg of protein from each lysate was denatured, dissolved and separated with sodium dodecyl sulphate-polyacrylamide using gel electrophoresis (SDS-PAGE) in an XCell4 SureLock™ Midi-Cell system (Invitrogen, Carlsbad, CA, USA). The separated proteins were transferred to a polyvinylidene fluoride (PVDF) or nitrocellulose membrane and blocked with 5% milk in Tris-buffered saline with Tween-20 (TBST) for 1 h at room temperature. After washing three times with TBST, the membrane was probed with the primary antibodies corresponding to the proteins of interest in TBST overnight at 4 °C. The following day, the fluorescent secondary antibody was applied in TBST for 1 h at room temperature. 

### 4.8. Statistical Analysis

All data are expressed as mean ± SEM of 6 individual experiments in each study group. The results for the cell glucose uptake are presented as the mean percentage (%) from the value of the first experiment in the glucose control group. This procedure provided a statistic (SEM) for the spreading of the dataset (including control) over day-to-day experimental runs. Two-way repeated-measures analysis of variance (ANOVA) was applied to identify the treatment effects. When a significant F-ratio was obtained, a least significant difference (LSD) post-hoc test was used to locate specific between-treatment differences. Significance was set at the *p <* 0.05 level of confidence.

## Figures and Tables

**Figure 1 ijms-21-05942-f001:**
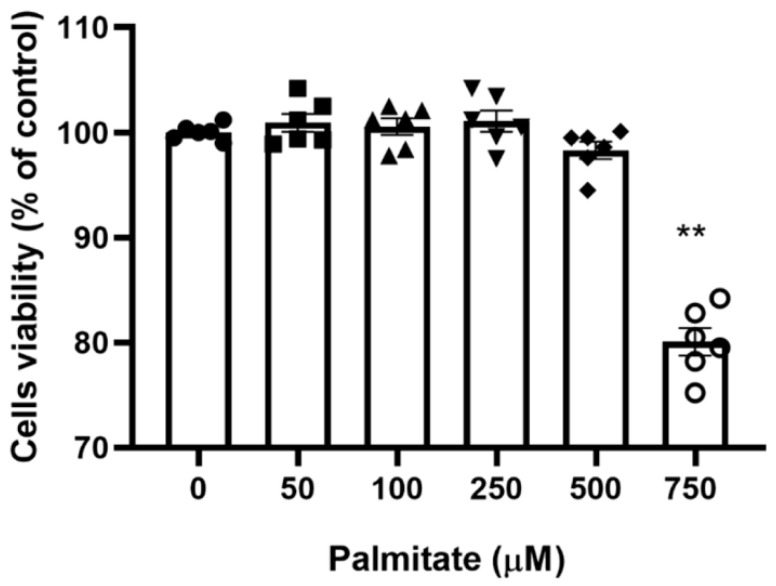
C2C12 myotubes viability with increasing concentration of palmitate. C2C12 myotubes were treated with increasing palmitate concentrations from 0 (control) to 750 μM; (*n* = 6). ** Significantly different from control (*p <* 0.01). Data represent mean ± SEM of 6 individual experiments in each study group.

**Figure 2 ijms-21-05942-f002:**
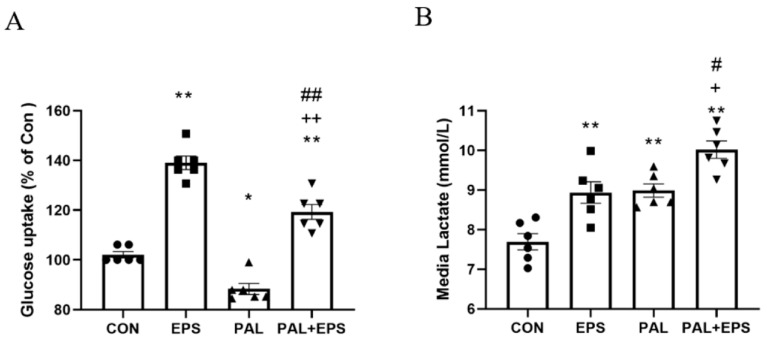
C2C12 myotube glucose uptake (**A**) and cell media lactate concentration (**B**) after exposure to either standard growth media (CON), 16 h of electrical pulse stimulation (EPS; 11.5 V, 1 Hz, 2 ms), 300 μM palmitate (PAL) or the combination of both palmitate and EPS (PAL + EPS). Significantly different from control (CON, * *p <* 0.05, ** *p <* 0.01). Significantly different from palmitate (PAL; ^#^
*p* < 0.05, ^##^
*p <* 0.01). Significantly different from EPS (^+^
*p <* 0.05, ^++^
*p <* 0.01).

**Figure 3 ijms-21-05942-f003:**
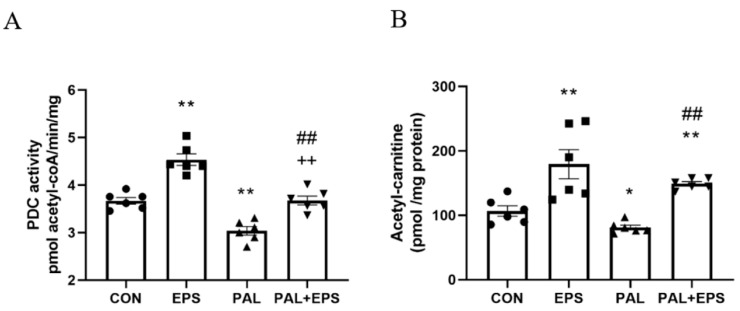
C2C12 myotube pyruvate dehydrogenase complex (PDC) activity (**A**) and acetylcarnitine content (**B**) after exposure to either standard growth media (CON), 16 h of electrical pulse stimulation (EPS; 11.5 V, 1 Hz, 2 ms), 300 μM palmitate (PAL) or the combination of both interventions (PAL + EPS). Significantly different from control (CON; * *p <* 0.05, ** *p <* 0.01). Significantly different from palmitate (PAL; ^##^
*p <* 0.01). Significantly different from EPS (^++^
*p <* 0.01). Data represent mean ± SEM of 6 individual experiments in each study group.

**Figure 4 ijms-21-05942-f004:**
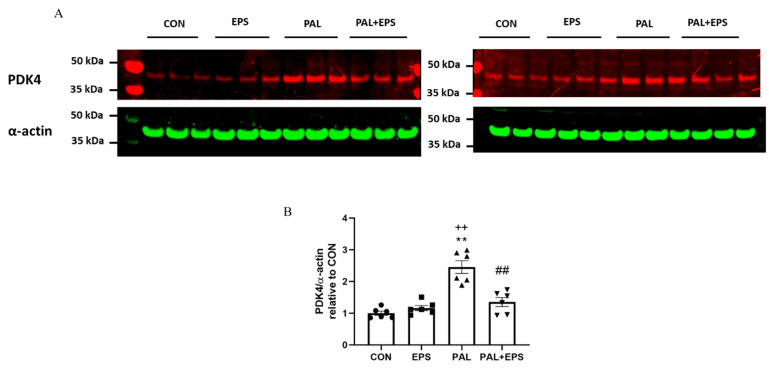
Western blots depicting pyruvate dehydrogenase kinase isoform 4 (PDK4) and α-actin protein bands with their corresponding molecular weights (**A**), their mean band intensities normalised to α-actin (**B**) in C2C12 myotube after exposure to either standard growth media (CON), 16 h of electrical pulse stimulation (EPS; 11.5 V, 1 Hz, 2 ms), 300 μM palmitate (PAL) or the combination of both interventions (PAL + EPS). Significantly different from control (CON; ** *p <* 0.01). Significantly different palmitate (PAL; ^##^
*p <* 0.01). Significantly different from EPS (^++^
*p <* 0.01). Data represent mean ± SEM of 6 individual experiments in each study group.

**Figure 5 ijms-21-05942-f005:**
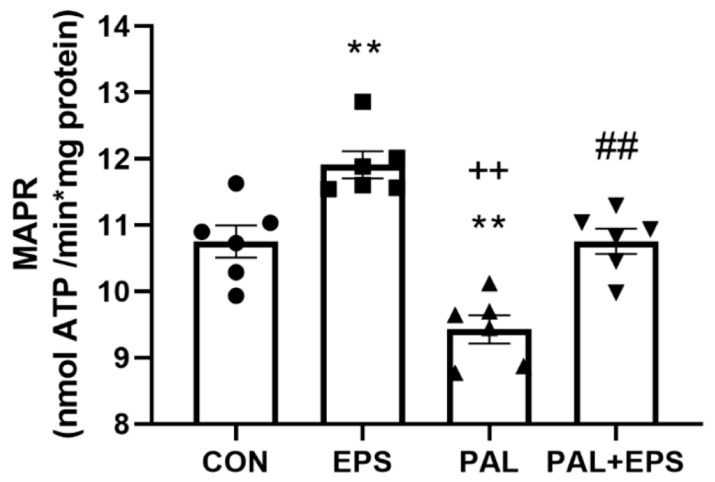
C2C12 myotube maximal mitochondrial ATP production rates (MAPR) from pyruvate + malate after exposure to either standard growth media (CON), 16 h of electrical pulse stimulation (EPS; 11.5 V, 1 Hz, 2 ms), 300 μM palmitate (PAL) or the combination of both interventions (PAL + EPS). Significantly different from control (CON, ** *p <* 0.01). Significantly different from palmitate (PAL, ^##^
*p <* 0.01). Significantly different from EPS (^++^
*p <* 0.05). Data represent mean ± SEM of 6 individual experiments in each study group.

**Figure 6 ijms-21-05942-f006:**
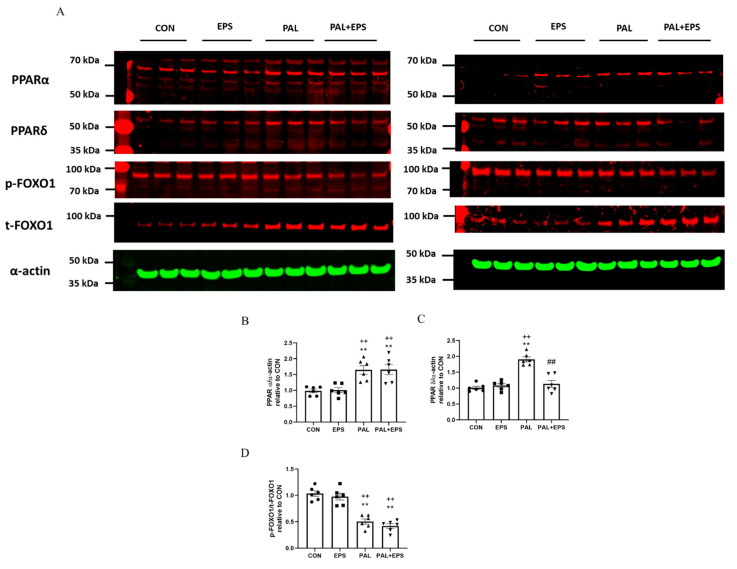
Western blots depicting PPARα, PPAδ, phosphorylated (*p*)-FOXO1, total (t)-FOXO1 and α-actin protein bands with their corresponding molecular weights (**A**), mean band intensities normalised to α-actin for peroxisome proliferator-activated receptor alpha (PPARα) (**B**), peroxisome proliferator-activated receptor delta (PPARδ) (**C**), and *p*/t-FOXO1 (**D**) protein expression in C2C12 myotubes whole cell lysates after myotube exposure to either standard growth media (CON), 16 h of electrical pulse stimulation (EPS; 11.5 V, 1 Hz, 2 ms), 300 μM palmitate (PAL) or the combination of both interventions (PAL + EPS). Significantly different from control (CON; ** *p <* 0.01). Significantly different from palmitate (PAL; ^##^
*p <* 0.01). Significantly different from EPS (^++^
*p <* 0.01). Data represent mean ± SEM of 6 individual experiments in each study group.

**Figure 7 ijms-21-05942-f007:**
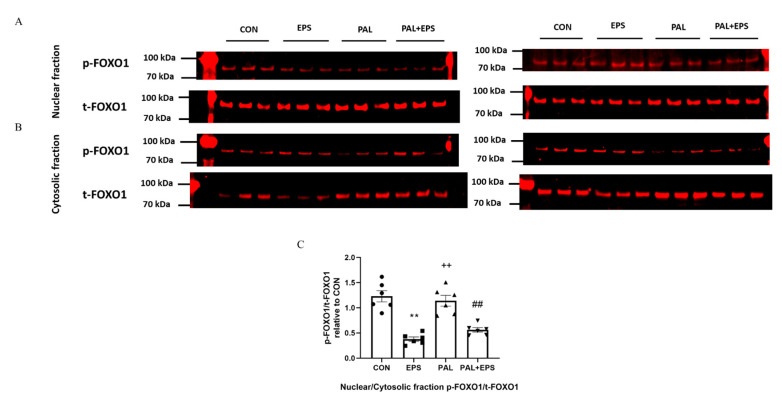
*p*- and t-FOXO1 protein expression in nuclear (**A**) and cytosolic (**B**) cell fractions and their *p*-FOXO1/t-FOXO1 ratios normalised to that of nuclear to cytoplasmic factor (**C**) in C2C12 myotubes after exposure to either standard growth media (CON), 16 h of electrical pulse stimulation (EPS; 11.5 V, 1 Hz, 2 ms), 300 μM palmitate (PAL) or the combination of both interventions (PAL + EPS). Significantly different from control (CON; ** *p <* 0.01). Significantly different from palmitate (PAL; ^##^
*p <* 0.01). Significantly different from EPS (^++^
*p <* 0.01). Data represent mean ± SEM of 6 individual experiments in each study group.

**Figure 8 ijms-21-05942-f008:**
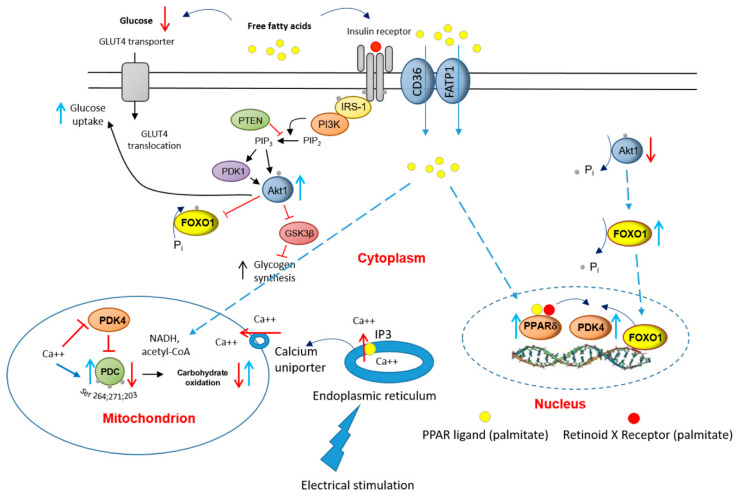
A schematic illustration of the underlying mechanism by which palmitate inhibits the carbohydrate (CHO)/glucose-derived pyruvate mitochondrial ATP production through activation of PPARδ- and FOXO1-mediated upregulation of PDK4 protein. Blue upright arrow denotes upregulation and red downward arrow denotes downregulation. CD36, fatty acid translocase; FATP1 insulin-sensitive fatty acid transporter; PDK1, 3-phosphoinositide-dependent kinase-1; GLUT4, glucose transporter isoform 4; IGF, insulin-like growth factor; FFA, free fatty acid; Akt1, serine/threonine-protein kinase 1; PTEN, phosphatase and tensin homolog; PI3K, phosphoinositol 3-kinase; GSK3β, glycogen synthase kinase; IRS-1, insulin receptor substrate 1.

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
