# Peer review of "PPARδ and FOXO1 Mediate Palmitate-Induced Inhibition of Muscle Pyruvate Dehydrogenase Complex and CHO Oxidation, Events Reversed by Electrical Pulse Stimulation"

_ijms, 2020, doi:10.3390/ijms21165942_

Round 1
Reviewer 1 Report
The authors have performed invitro analysis of C2C12 myotubes to answer how palmitate perturbs the PDC controlled glucose oxidation. The study is well performed. I have minor concerns listed below.
In Fig 3, the authors show that EPS could rescue Palmitate induced PDC activity where control seems comparable to rescue. However, in rescue, Acetyl carnitine is elevated compared to control. The authors should discuss this result with possible explanation. Is the experiment sufficiently powered?
Fig 4, Fig 6 and Fig 7: Please provide the full western blot. The western blot data represents 3 samples instead of 6. Please provide the data for all the samples. Actin is used as a control in this case. The authors should provide evidence that this protein does not change under the effect of palmitate/EPS in muscle myotubes.
The authors should comment on the status of Ca ions under the EPS condition.
Why did authors choose to measure only PPAR alpha and delta? Why were the other PPARs not quantified?
Line 188: Typographical error (PPAR..)
Authors should write an explanation as to why the decrease in glucose uptake in Palmitate group is accompanied by high lactate in media. A low glucose uptake should result in low lactate in principle.
Can the authors support this mechanistic evidence in in-vivo model?
Reviewer 2 Report
Chien et al. investigated the effect of electrical pulse stimulation on pyruvate dehydrogenase complex activity and carbohydrate oxidation in myotubes treated with palmitate. The manuscript is interesting and potentially important. My comments are relatively minor.
- Fig 6A. Bands of PPARα and PPARδ are not clear. Please change to more obvious pictures.
- P9 line262, what does pig mean?
- P10 line 300, the tone of the conclusion should be lowered. Because knockdown or inhibition of PPARδ and FOXO1 had not been performed in this manuscript, it is difficult to mention “a central role for PPARδ and FOXO1 in the palmitate induced …”
- P12 Line353, cell extract lactate? Not medium lactate concentration?
- No information about duration of palmitate exposure. Please indicate when palmitate exposure started and how long exposed.
- P12 line 344, citing Ref#40 is probably wrong. Glucose uptake data from Ref#40 is during hyperinsulinaemic-euglycaemic clamp, but data in this manuscript is basal, not insulin stimulated state. Please cite an adequate reference. In addition, was glucose transport activity measured for 15 min immediately after 16hrs EPS? Need a detail of experiment.
- Is there any evidence that palmitate exposure reduces contraction induced an increase in glucose uptake in muscle?
- What percentage of myotubes were contracted by EPS? Please indicate the percentage in method section.
Round 2
Reviewer 2 Report
good work.